# The Potential for the Direct and Alternating Current-Driven Electrospinning of Polyamides

**DOI:** 10.3390/nano12040665

**Published:** 2022-02-16

**Authors:** Pavel Holec, Radek Jirkovec, Tomáš Kalous, Ondřej Baťka, Jiří Brožek, Jiří Chvojka

**Affiliations:** 1Department of Nonwovens and Nanofibrous Materials, Faculty of Textile Engineering, Technical University of Liberec, 461 17 Liberec, Czech Republic; radek.jirkovec@tul.cz (R.J.); tomas.kalous@tul.cz (T.K.); jiri.chvojka@tul.cz (J.C.); 2Department of Textile Machines, Faculty of Mechanical Engineering, Technical University of Liberec, 461 17 Liberec, Czech Republic; ondrej.batka@tul.cz; 3Department of Polymers, Faculty of Chemical Technology, University of Chemistry and Technology Prague, 166 28 Praha, Czech Republic; jiri.brozek@vscht.cz

**Keywords:** nanofibers, polyamide, nylon, direct current (DC), alternating current (AC), electrospinning, polymer solution

## Abstract

The paper provides a description of the potential for the direct current- and alternating current-driven electrospinning of various linear aliphatic polyamides (PA). Sets with increasing concentrations of selected PAs were dissolved in a mixture of formic acid and dichloromethane at a weight ratio of 1:1 and spun using a bar electrode applying direct and alternating high voltage. The solubility and spinnability of the polyamides were investigated and scanning electron microscopy (SEM) images were acquired of the resulting nanofiber layers. The various defects of the spun fibers and their diameters were detected and subsequently measured. Moreover, the dynamic viscosity and conductivity were also subjected to detailed investigation. The most suitable concentrations for each of the PAs were determined according to previous findings, and the solutions were spun using a Nanospider^TM^ device at the larger scale. The fiber diameters of these samples were also measured. Finally, the surface energy of the fiber layers produced by the Nanospider^TM^ device was measured aimed at selecting a suitable PA for a particular application.

## 1. Introduction

Aliphatic polyamides (also known as nylons) are widely-used thermoplastic polymers [1]. They are synthesized via the polycondensation of an Ω-aminoacid, the polyreaction of a diamine with a dicarboxylic acid or halide and the ring-opening polymerization reactions of lactams [2]. The abbreviations of the various nylons consist of the prefix PA (polyamide) plus numbers that identify the number of carbon atoms between adjacent amide groups. A PA synthesized from a diamine and a dicarboxylic acid (or halide) is identified by two numbers–the first corresponds to the number of carbons between the diamine groups and the second to the number of carbons in the whole of the dicarboxylic acid (i.e., halide) [3]. The polar amide groups allow for secondary interactions, including hydrogen bonding. Consequently, PAs usually comprise semicrystalline polymers with good mechanical properties. Simultaneously, the aliphatic groups serve to render the whole of the polymeric chain more flexible.

Their ability to interact with water means that nylons are affected by moisture, which can change their mechanical and dimensional properties [4]. The properties of the various types of nylon depend on the number of amide groups in the macromolecule. More polar nylons usually exhibit improved mechanical strength properties due to the formation of linkages between the amidic groups. The moisture absorption ability and crystallinity are enhanced with the increasing polarity of the nylons. Conversely, the specific gravity decreases [5]. The range of types of nylons allows for the gradual modulation of their properties, while the chemical basis remains similar. No other mass-produced polymers offer this advantage. Due to their good mechanical and chemical properties, nylons are used in the automotive [6], textile [7], engineering, high-performance plastics [8], coating, packaging [9] and health [10] sectors. Polyamides, with the exception of polyamide 4, which undergoes degradation near the melting point [11], can be processed by conventional technologies such as injection molding and spinning.

The demand for low diameter polymeric fiber products, especially for filtration and medicinal applications has over the last 20 years increasingly focused attention on nanofibers, which evince unique properties. The diameters of such fibers usually range below 1000 nm. According to some scientific or political organizations nanomaterials and nanoscale refers to the range of approximately 100 nm to 1000 nm. This adjustment was implemented to prevent single atoms or their small groups from being included in the nanoscale dimension [12]. The most characteristic properties of nanofibers is their high surface ratio compared to their volume [13]. This makes them ideal for use as membranes for air and water, chemical catalysis and medical purposes, e.g., as scaffolds for tissue engineering, wound dressing and drug delivery systems, and as fuel and battery membranes [14]. In addition, nanofibers can be further functionalized via additives, coatings, chemical grafting, crosslinking, annealing and plasma treatment [15].

One of the approaches to producing nanofibers comprises the use of high-voltage polymeric solutions [16,17]. This electrohydrodynamic phenomenon results in the creation of so-called Tylor cones on the deformed surface of the solution. Subsequently jets are formed at the top of the cones. After that, they are elongated and their diameter gradually decreases leading to the creation of nanofibers. The fibers can then be collected and further processed [18]. The electrospinning process can be driven by direct current (DC) or alternating current (AC). To date, the application of the DC electrospinning process has been the preferred option. The first technology for the large-scale production of nanofibers, the Nanospider^TM^ (Elmarco, Liberec, Czech Republic), used a DC energy source (based on the respective patent [19]). AC electrospinning, however, is a more recently-developed method, concerning which research to date has been significantly less extensive than that of the DC process [20]. AC driven electrospinning differs from the DC version mainly in the rapid changing of charges and discharging of produced fibers. Where Tylor cones and jets created via DC power source can be stable and thus DC electrospinning could be continual, the AC-driven electrospinning undergoes a periodical changes of charges. In this case, cones and jets mostly rise and collapse in each half-wave of the voltage, and thus the spinning process is unsettled. Additionally, fibers spun via AC systems carry their charge for only a short amount of time because they are discharged by the following fibers created by the opposite charge. The final fiber structure carries almost no free charge and acts as an electroneutral material. Consequently the products created via the DC and AC approaches generally differ. While DC electrospinning produces thin, relatively homogeneous compact layers, the AC approach produces structures known as fiber plumes which are carried to the collector (which need not be electrically active) by a so-called ionic wind. Hence, the final product is more airy and bulky than DC-produced fibers [21]. The structure of the material influences its subsequent application. There are only a few articles comparing properties of DC and AC driven electrospun materials. It was shown on polycaprolactone solutions, that the used electrospinning technology can change the morphology and thus surface properties of nanofiber layers [22].

Nylon polymers are used widely in the electrospinning production of nanofibers. The most commonly used polymers is PA6, followed by PA6|6, PA11 and PA12, respectively. The main applications of nylon nanofibers comprise filtration [23], solid-phase extraction [24], wound dressings [25], tissue engineering [26] and electrotechnology [27]. Formic acid is the most common solvent for PA6 and PA6|6, sometimes in mixtures with acetic acid, dichloromethane or chlorophenol [28]. According to the solvent used, several morphological and structural variants may appear in electrospun PA samples such as ribbons, branches, bending loops, shrinkages, nanosheets or high-aspect ratio nanofibers [28,29,30,31]. They are affected not only by the solvent but also by the concentration and voltage applied. However, since nylons and their copolymers with longer alkane chains do not dissolve well in such solvent systems, more expensive and ecologically-unfriendly solvents e.g., 2,2,2-trifluoroethanol [32], 1,1,1,3,3,3-hexafluoropropan-2-ol or cresol are required [33], although, in this connection, a number of papers have suggested the dissolution of PA11 solely in formic acid [34,35]. Behler et al. demonstrated in 2007 that PA11 and PA12 can be dissolved in a mixture of formic acid and dichloromethane at a 1:1 weight ratio and subsequently electrospun [33]. This solvent can also be applied for dissolving even the most polar nylons, and thus, provides a suitable solvent for the comparative research of the spinnability of various PAs.

## 2. Materials and Methods

### 2.1. Materials and Solution Preparation

Several types of polyamides (PA) were tested: Two PA4 variants were synthesized at the Department of Polymers, UCT Prague (according to [11]) with differing viscosity average molecular weights (M_v_) of 18,000 and 280,000 g·mol^−1^. PA4|6 (Stanyl^®^ HGR3-W) was obtained from the DSM company (Emmen, the Netherlands), PA6 (Ultramide^®^ B72) with a weight average molecular weight (M_w_) of 21,050 g·mol^−1^ and a dispersity Đ = 3.2 was obtained from BASF (Ludwigshafen, Germany). PA6|6 (CAS 32131-17-2) and PA6|12 (CAS 26098-55-5) were purchased from Sigma Aldrich (Praha, Czech Republic) and PA6|10 (Mw 1500 g·mol^−1^, CAS 9011-52-3), PA11 (CAS 25587-80-8) and PA12 (CAS 25038-74-8) from Scientific Polymer Production (New York, NY, USA). Formic acid (p.a., CAS 64-18-6) and dichloromethane (p.a., CAS 75-09-2) were obtained from Penta (Praha, Czech Republic).

Each type of polyamide was dissolved in a mixture of formic acid and dichloromethane at a weight ratio of 1:1 and stirred using a magnetic stirrer for 24 h at room temperature in closed containers. All the measurements of the solutions and electrospinning were conducted immediately following stirring.

### 2.2. High Voltage System and Electrode Geometry

A 100 mm-long bar electrode with a diameter of 10 mm was used for electrospinning which was driven either via direct (DC) or alternating (AC) current. The grounded collector comprised a 1 mm-thick steel plate with dimensions of 250 × 250 mm^2^ with rounded edges. The distance from the top of the electrode was set at 100 mm for the DC experiments. The electrospinning process usually lasted 60 s unless the process stopped spontaneously. The AC experiments were conducted without an electrically-active collector. The geometry of the spinning system was designed as symmetrical to the axis of the electrode. The geometry and the corresponding 2D electric field model of the DC experiments are shown in Figure 1. The model was created using the Autodesk Simulation Mechanical 2015 FEM software (version 2015.20, San Rafael, CA, USA). The task was simulated as 2D axisymmetric, where the model axis corresponded to the electrode axis. The shape of the polymeric solution droplet was based on an ellipse with a height of 4 mm. The assembly was enclosed in a box with a diameter of 300 mm and a height of 500 mm. The distance between the lower edge of the pad and the lower edge of the box was 144 mm. A voltage of 30 kV was set for the electrode and 0 V for the collector. The box’s boundaries were assumed to be perfectly isolated from the surrounding vicinity.

The direct current for the testing of the spinnability of the polyamides was provided by an AU-60P0.5-L (Matsusada Precision, Ōtsu, Japan) power source and the alternating current via a self-made HV source comprising an KGUG 36 transformer (ABB, Praha, Czech Republic) equipped with a ESS 104 regulator (Thalheimer-Trafowerke, Ostrava, Czech Republic) operated at a frequency of 50 Hz. 30 kV was applied for the DC electrospinning and an effective 30 kV for the AC electrospinning. Thus the total amount of energy was the same for both sources. Simultaneously, the certainty of exceeding the critical voltage for both electrospinning systems was ensured. The final spinning of the highest-quality resulting materials was conducted using an Elmarco Nanospider^TM^ NS 1WS500U device with a 50 kV spinning electrode and a −10 kV collector with a distance between them of 170 mm. The spinning of all the samples was performed at room temperature and at a relative humidity of approximately 40%.

### 2.3. Scanning Electron Microscopy Analysis

The electrospun nanofibers were coated with a 10 nm layer of gold using a Q150R ES rotary pumped coater (Quorum, Lewes, UK). The morphological analyses of the samples were obtained using a Vega 3 scanning electron microscope (TESCAN, Brno, Czech Republic) at an accelerated voltage of 20 kV. The diameters of the nanofibers were measured using ImageJ software (version 1.52a, Bethesda, MD, USA), and Veusz software (version 3.3.1, Garching, Germany) was used to create the corresponding histograms and graphs.

### 2.4. Viscometry Analysis

The viscometry analysis was conducted using a HAAKE Rotovisco (Thermo Fisher Scientific, Praha, Czech Republic) rheometer with a C35/1°Ti L cone and gap distance of 0.1 mm. The shear rate was linearly increasing from 300 to 3000 within 30 s. Each sample was tested five times at a temperature of 23 °C. The concentration dependences were evaluated by comparing the dynamic viscosities of the samples at shear rate of 500 s^−1^.

### 2.5. Conductivity Analysis

A Eutech Instruments CON 510 (Thermo Scientific Eutech Instruments, Landsmeer, The Netherlands) with a K10/6MM8 probe was used for the conductivity measurements. Each sample was tested three times at 21 °C.

### 2.6. Contact Angle and Surface Energy of the Fiber Layers

The contact angle was measured using a See System E instrument (Advex Instruments, Brno, Czech Republic). Glycerol was used to measure the contact angle; 5 µL was dosed in all cases onto the prepared fiber layers. A total of 30 contact angle measurements were taken for each sample. The surface energy was calculated via the Kwok-Neumann model (Equation (1)):(1)$$\gamma_{sl}=\gamma_{lv}+\gamma_{sv}-2\sqrt{\gamma_{lv}\gamma_{sv}}\cdot \left(1-0.0001057{\left(\gamma_{lv}-\gamma_{sv}\right)}^{2}\right),$$
where *γ_sl_*, *γ_lv_* and *γ_sv_* represent interfacial tensions of the solid-liquid, liquid-vapor and solid-vapor respectively.

## 3. Results

### 3.1. Material and Solution Preparation

Sets of five or six solutions with increasing concentrations were prepared for all the types of PA. The number of solutions of each polyamide depended on the solubility of the respective PA. All the tested PA types were found to be soluble in formic acid and dichloromethane mixed solvent. However, the solubilities of the PA11 and PA12 were limited to approximately 12% wt. and 10% wt., respectively. These solutions were prone to rapid precipitation on their surfaces which led to the formation of thin polymeric films, which acted to halt the spinning process. The PA12 was more prone to this behaviour than the PA11. This surface precipitation was caused by the preemptive evaporation of dichloromethane from the polymeric solution. The solubility of the PAs decreased with the decreasing polarity of the macromolecule. We assumed that increasing the content of dichloromethane in the solvents used for polyamides with longer aliphatic chains (11 and more) might improve the spinning process. Partial modification of the solvent system for PAs with aliphatic chains lower than 11 could be done by substituting dichloromethane with tetrachloroethane or their suitable mixture. This lowers the solvent’s evaporation rate and allows better controllability of the spinning process and the solution storage. On the other hand, this solvent system would probably not be suitable for PA11 and PA12, thus limiting the general applicability of the solvent for polyamides.

### 3.2. DC and AC Electrospinning

Although it was possible to electrospin most of the prepared solutions, the spinning process was not always ideal and, in such cases, would be considered inappropriate for application at the higher fabrication scale. For this reason, the spinnability of each sample was estimated both macroscopically and microscopically. Furthermore, not all the required values could be obtained for all the prepared solutions due to the inadequate spinning properties—an insufficient level of spinning efficiency, the spinning of fibers with too large diameters or the inability to spin at all. The DC macroscopic aspect served for the optical evaluation of the spinning process and the spun layers, i.e., their proportions and homogeneity. The most homogeneous and compact layers were evaluated as being of the highest quality. Generally, the layers prepared from the lower concentration solutions were less distinctive but more compact. Increasing amounts of polymer in the solutions allowed the formation of separated bolder fiber deposits. This effect is shown in Figure 2a. Further increases in the concentration led to a reduction in the effectiveness of the spinning process or its complete cessation.

The macroscopic evaluation of the AC electrospinning process was more complicated than that of the of the DC process due to the differing character of the AC spun fibers (Figure 2b). The dependence of spinnability on the concentration was not as apparent as it was for the DC process. The AC spun fibers in the form of so-called fiber plumes were carried via the ionic wind to the electroneutral collector. The resulting bulky and sticky product created non-homogeneous formations, which were difficult to evaluate optically. Hence, those samples with the most significant volume were considered the most suitable. The optimal concentrations of the most favorable spinning results were usually related to the DC concentrations. However, in some cases, the concentrated solutions were found to spin more efficiently (PA6|6, PA6|10 and PA11). Table 1 provides the results of the macroscopic observation. It is clear from the table that in the case of PA4 samples there is an order of magnitude difference in the concentration the formation of continuous fiber layers. The concentration of solution is in relation of the entanglements formation which depends on molar mass of PA and the solvent quality.

The SEM images of the DC electrospun samples revealed a tendency toward the creation of both fibers and ribbons. Mixtures of fibers and ribbons were observed in the PA4|6 (Figure 3a), PA6, PA6|6, PA6|10, PA6|12 and PA11 samples. The ribbons appeared more frequently in the higher concentration solutions than in the optimal solutions. Some of the solutions created ribbon structures only. No ribbons were created at even higher concentrations, and wrinkled microfibers were discovered in the PA4 (280,000), PA6|6, PA11 and PA12 samples.

The significant creation of ribbons was not observed for the AC electrospun samples (Figure 3b). However, it is possible that the identification of ribbons in the AC samples was more complex than for the DC samples. The DC samples were forced toward the collector and the final product was deposited in the form of compacted layers; hence the flat faces of the ribbons were easily recognizable. On the other hand, the AC approach produced bulky structures so that the ribbons were distinguishable only in those positions where they were twisted or bent.

The diameters of the fibers and the standard deviations increased with the increasing polymer concentration in the solution (Figure 4). The increase in the standard deviations was caused mainly by the increasing numbers of ribbons in the samples. In order to obtain reproducible data, only the broadest ribbon diameters were measured, which resulted in an increase in the difference between the standard fibers and the ribbons. The diameters of the ribbons increased with the concentration only slowly.

Figure 4a,b illustrate that the narrowest fiber diameters were usually obtained from those solutions with the lowest concentrations. This was observed for both the DC and the AC spinning systems. It was usually easier to prepare fibers of lower diameters from the lowest concentration solutions via DC spinning rather than via AC spinning. Further, the AC system was unable to sufficiently spin some of the most concentrated solutions (16% PA4|6, 14% and 16% PA6|6).

### 3.3. Viscometry and Conductivity Analysis

Figure 5a illustrates the increasing dynamic viscosities of the PA solutions with their increasing weight concentrations. High molecular weights led to the high viscosity values of the PA4 280,000, as well as, presumably, the PA6|6. The conductivity of the primary solvent was 20.5 ± 0.3 µS·cm^−1^. In most cases, the conductivity of the PA solutions increased or showed no major changes with increasing concentration (Figure 5b). Only the conductivity of the PA4 18,000 was observed to decrease with the concentration, which was due to the to the contribution of end groups of polymer with low molar mass.

### 3.4. Nanospider DC Spinning

The optimal DC electrospinning concentration for each type of PA was chosen after considering the macroscopic and microscopic spinning aspects and the corresponding fiber diameters. Since problems arose with the drying of the 6% PA11 on the string electrode, the solution concentration of the PA11 was lowered slightly to 5.5%, which allowed for the long-term spinning of a homogeneous layer. Only the PA12 failed to demonstrate good electrospinning properties from the applied solutions-the drying of the polymer on the string electrode was so rapid that even lowering the concentration was insufficient. Since it was difficult to fabricate nanofibrous layers of the required quality, the PA12 was not spun using the Nanospider^TM^ machine. The optimal concentrations of the successfully spun PA are shown in Figure 6 with the corresponding fiber diameter histograms, which were created using 500 measured diameters for each sample.

### 3.5. Contact Angle and Surface Energy

The surface energy and the contact angle were measured so as to allow for the selection of a suitable PA for the desired application, for which wetting is an important property. The contact angle was measured using glycerol, and the surface energy was then calculated from the contact angle. The resulting contact angle and surface energy values are shown in Figure 7.

The results revealed that higher numbers of carbons in repeating unit of polyamide lead to larger contact angles and reductions in the surface energy. In the case of low molecular weight PA 4 (18,000), the values of surface wettability and surface energy are out of the stated dependence for polyamides. The differences are due to a higher proportion of terminal polar amino groups. PA 4 (18,000) exhibited a high surface energy (60.88 mJ·m^−2^) and was, thus, more wettable than, for example, PA 11, which had a lower surface energy (47.77 mJ·m^−2^). Therefore, by selecting a suitable PA, it is possible to select the required wettability of the nanofiber layer.

## 4. Limitations of the Study

It is important to bear in mind a number of limitations with respect to the presented experiments and their results. Firstly, the tested PAs were not of identical molecular weights and dispersities; in some cases, it was difficult to obtain accurate information on these two parameters, even from the manufacturers and sellers of the polymers. Secondly, the concentration increments (usually of 2% wt.) aimed at determining the best concentrations for the DC and AC electrospinning processes were relatively large. Hence, the presented “optimal” concentrations corresponded to the tested concentrations, which may not have been the most appropriate on a global scale. Thirdly, the bar electrode used only small amounts of the static polymeric solution. Therefore, the dependence of the spinning system on the solution and changes in the atmospheric conditions may have affected the spinning process. In addition, the spinning time was limited.

## 5. Conclusions

This paper addressed the potential for the electrospinning of PA4, PA4|6, PA6, PA6|6, PA6|10, PA6|12, PA11 and PA12 solutions in a formic acid and dichloromethane. The solubility decreased with the increasing length of the aliphatic chain in repeating structure unit. Hence the solubilities of PA11 and PA12 were limited to 12 % wt. and 10 % wt. respectively. The ideal concentrations for both the DC and AC electrospinning techniques for all the tested PA solutions were determined by comparing the macroscopic behavior of the spinning processes from the bar electrode and the microscopic study of the resulting structures. The macroscopic aspect of the spinning process helped to identify the efficient and non-efficient processes, and the microscopic analysis was useful in determination of the quality of the prepared fibers and to identify any defects present. The creation of ribbons from the more concentrated solution was identified when applying the DC spinning technique. The application of the AC system resulted in the occurrence of a minimal amount of ribbons. Increases in the fiber diameters with the increasing concentration of the spinning solution were determined for all the solutions. However, no clear correlation was discovered between the DC and AC fiber diameters, i.e., the two spinning systems produced fiber layers with similar qualities.

The viscosities increased with the concentration of the solutions. The conductivities usually increased to a value of approximately 1500 μS·cm^−1^ for all the PA types except for PA4 (18,000), the conductivity of which decreased with rising concentration due to an excess of macromolecules that obstructed the charged particles.

Finally, the PA solutions with the optimal concentrations were electrospun using a Nanospider^TM^ device, which demonstrated that the used solvent was suitable for the DC electrospinning of various types of linear non-aromatic PAs. The nanofibrous layers obtained were of a sufficient quality and had no defects. The consecutive measurements approved that the decreasing polarity of nylons tends to increase contact angle and decrease the surface energy of the corresponding nanofiber layers. Hence the PA nanofiber layers of similar chemical structures can provide different surface properties. It might help choose the most suitable polyamide as a material for filtering specific liquids and producing polyamide nanofiber coats or composites, where proper surface interaction is essential for their manufacturability, mechanical properties and durability.

The use of a single solvent for a large number of PAs will allow for the conducting of a wide range of further research into the spinning of PAs and the DC and AC spinning of PA mixtures thereof. In addition, the findings will allow for the simplification of the investigation of the influence of structure of macromolecules on the spinning process via the use of a single solution for a variety of similar polymers.

## Figures and Tables

**Figure 1 nanomaterials-12-00665-f001:**
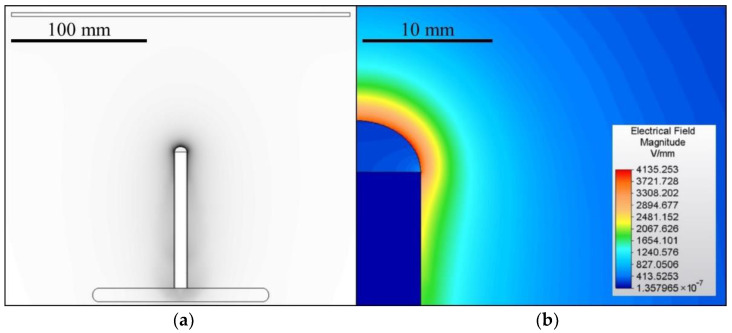
The geometry of the DC electrospinning system (**a**) and a detail of the corresponding electrostatic field (**b**). The spinning system for the AC tests applied the same geometry as the DC system but without a collector.

**Figure 2 nanomaterials-12-00665-f002:**
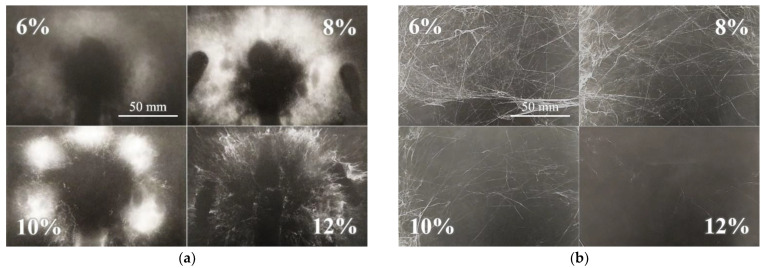
Macroscopic view of the DC electrospinning of PA11 (**a**) illustrates the increasing distinctness and collateral separation of the spun layers (the scale from the first image applies to all the images). Example of an AC electrospun sample (**b**)—PA6|6 exhibited a similar spinnability trend to the PA11 spun via DC technology.

**Figure 3 nanomaterials-12-00665-f003:**
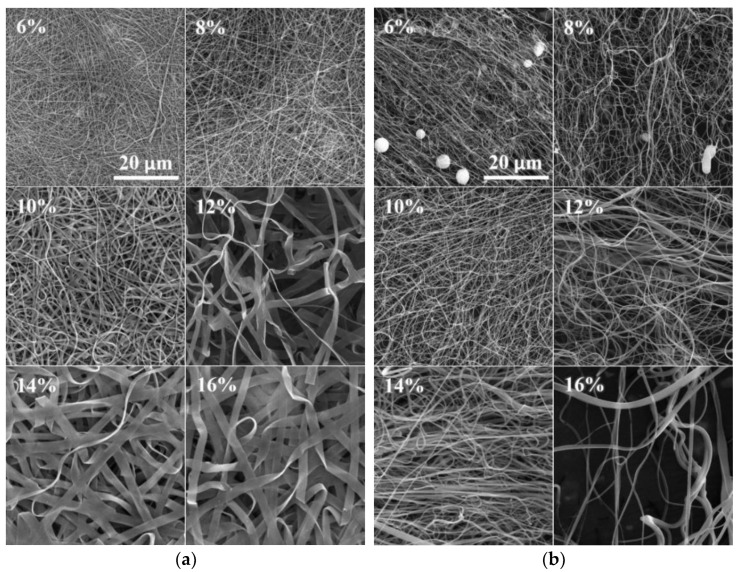
SEM images of the DC (**a**) and AC (**b**) electrospun PA4|6 nanofibers (the scale from the first image applies to all the images). Increasing numbers of ribbons were observed in the DC electrospun samples, whereas this phenomenon was not observed for the AC electrospun samples.

**Figure 4 nanomaterials-12-00665-f004:**
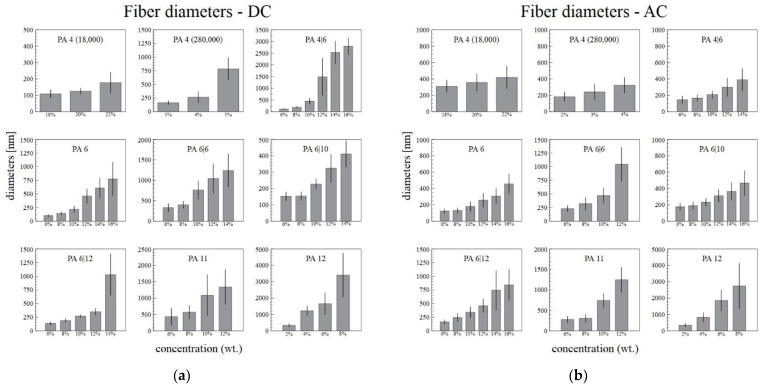
The dependence of the DC (**a**) and AC (**b**) electrospun fiber and ribbon diameters on the concentration of the solution, with the corresponding standard deviations.

**Figure 5 nanomaterials-12-00665-f005:**
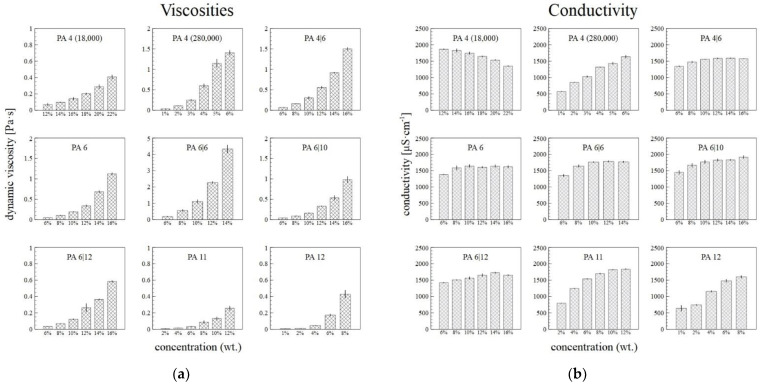
The dependence of the viscosity at shear rate of 500 s^−1^ (**a**) and conductivity (**b**) of the PA solutions on their concentrations, with the corresponding standard deviations.

**Figure 6 nanomaterials-12-00665-f006:**
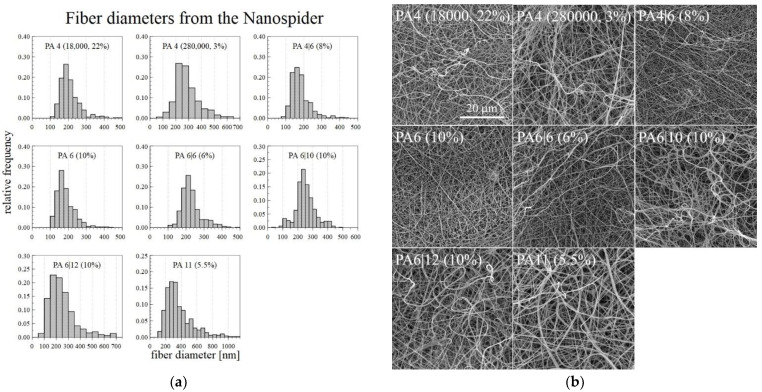
Histograms of the fiber diameters (**a**) and the corresponding SEM images (**b**) of the PA fiber layers prepared using the Nanospider^TM^ machine.

**Figure 7 nanomaterials-12-00665-f007:**
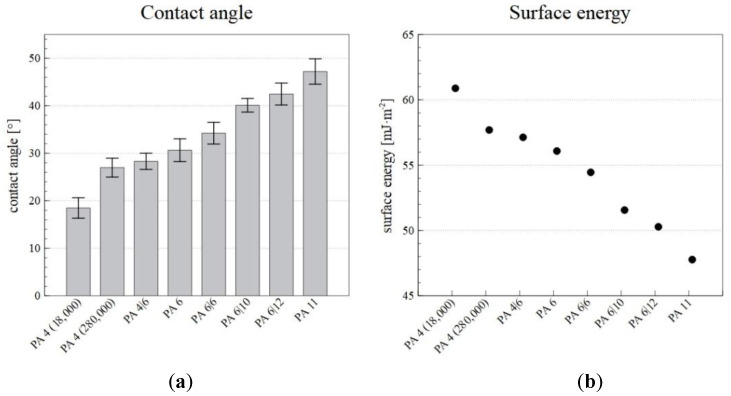
The contact angles (**a**) and surface energies (**b**) of the PA nanofiber layers produced via the Nanospider^TM^ device.

**Table 1 nanomaterials-12-00665-t001:** Results of the macroscopic observation of the spinning processes of the PA solutions showing the most suitable concentrations of the solutions used in both the DC and AC systems.

	PA4 (18,000)	PA4 (280,000)	PA4|6	PA6	PA6|6	PA6|10	PA6|12	PA11	PA12
DC	22%	3%	8%	10%	6%	10%	10%	6%	4%
AC	22%	3%	8%	10%	8%	12%	10%	8%	4%

## Data Availability

The data presented in this study is available on request from the corresponding author.

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
