# Peer review of "The Potential for the Direct and Alternating Current-Driven Electrospinning of Polyamides"

_nanomaterials, 2022, doi:10.3390/nano12040665_

Round 1

Reviewer 1 Report

The manuscrit reported the efforts on the DC and AC driven electrospinning of various polyamides. The current work provided much valuable information for the electrospinning preparation of high perfromance polymers, especaially the AC driven ones although many technical issues still exsited. It is quite instructive for the readers in the areas. Thus, it is my pleasure to suggest it to be accepted by the Journal with minor revisions.

  1. In the background part, the AC driven electrospinning procedure should be discussed more.
  2. Page 2, line 93, "cresole" should be "cresol";
  3. Page 5, line 173, since the low boiling point of dichloromethane casused the rapid envaporation of the solvent during ES procedure, the same series of solvents, such as tetrachloroethane with high bp might be beneficial for the preparation.  I think it is common the the all PA systmes besides PA11 and PA12. Could the authors discuss it in the revised manuscript?
  4. Effects of the DC and AC conditions on the micromorphologies of the derived PA fibers should be more explained so as to provide much  more valuable information for designing novel materials. 

Author Response

Author's Reply to the Reviewer 1 (Round 1)

Dear Reviewer 1,

Colleagues and I appreciate the revision of our paper. We tried to correct the text according to your suggestions in the following reply.

Sincerely,

Pavel Holec

  1. In the background part, the AC driven electrospinning procedure should be discussed more.

Our response:   We added a description of AC electrospinning and its comparison to the standard DC electrospinning technology to the Introduction section (lines 76-84 in the revised text). The text does not go into much detail. It aims to show basic information about both technologies and offer readers better insight to understand the concept of the paper better.

  1. Page 2, line 93, "cresole" should be "cresol";

Our response:   The cresole was rewritten to cresol.

  1. Page 5, line 173, since the low boiling point of dichloromethane casused the rapid envaporation of the solvent during ES procedure, the same series of solvents, such as tetrachloroethane with high bp might be beneficial for the preparation. I think it is common the the all PA systmes besides PA11 and PA12. Could the authors discuss it in the revised manuscript?

Our response:   The possibility of using tetrachloroethane in place of dichloromethane was introduced in the Material and solution preparation section (lines 193-198 in the revised text). The opportunity for further optimisation of lower aliphatic chains PAs solutions is presented.

  1. Effects of the DC and AC conditions on the micromorphologies of the derived PA fibers should be more explained so as to provide much more valuable information for designing novel materials.

Our response:   There is only a limited number of corresponding publications on the topic. Thus, we cannot evaluate for certainty the influence of both electrospinning methods on the micromorphology of the fibers. Therefore we add a mention (lines 89-92 in the revised text) that it was shown that a used electrospinning technology could change the properties of the nanofibrous layers. The statement is supported by reference 22 (in the new numbering). It is important to point out that the reference is considered as auto-citation.

Reviewer 2 Report

This paper shows the radiation study of polyamide using the Nanospider TM device and the properties of the emitted material. A few things to say: 1. It would be good if you indicate the country of origin of the materials and equipment used. 2. It would be good to be a little more specific about the applied research you wrote in the conclusion section. 3. Is there any reason you used glycerol to measure the PA contact angle? Additionally, it would be nice if there was an expression for calculating the surface energy through the contact angle. 

Author Response

Author's Reply to the Reviewer 2 (Round 1)

Dear Reviewer 2,

Colleagues and I appreciate the revision of our paper. We tried to correct the text according to your suggestions in the following reply.

Sincerely,

Pavel Holec

  1. It would be good if you indicate the country of origin of the materials and equipment used.

Our response:   All materials and equipment countries' origins were added to the Material and methods section.

  1. It would be good to be a little more specific about the applied research you wrote in the conclusion section.

Our response:   The Conclusion section was modified (lines 328-331 in the revised text) to be more specific concerning the paper's results application. We tried to avoid extensional references to other papers in this section because there are mentioned in the Introduction.

  1. Is there any reason you used glycerol to measure the PA contact angle? Additionally, it would be nice if there was an expression for calculating the surface energy through the contact angle.

Our response:   The equation of Kwok-Neumann model was added to the paper, and a corresponding reference was appended to the text (lines 177-179 in the revised text). It is important to mention here that the reference is considered auto-reference. There is no other paper comparing DC and AC electrospun nanofiber layers to our best knowledge. The glycerol was used for the measurement because See System E instrument software has well-defined glycerol parameters for the used Kwok-Neumann model.
